# Human Synovial Mesenchymal Stem Cells Expressed Immunoregulatory Factors IDO and TSG6 in a Context of Arthritis Mediated by Alphaviruses

**DOI:** 10.3390/ijms242115932

**Published:** 2023-11-03

**Authors:** Melissa Payet, Axelle Septembre-Malaterre, Philippe Gasque, Xavier Guillot

**Affiliations:** 1Research Unit ‘Etudes Pharmaco-Immunologiques’ UR EPI, Université de la Réunion, 97400 Saint-Denis, La Réunion, France; melissa.payet10@gmail.com (M.P.); axelle.malaterre-septembre@univ-reunion.fr (A.S.-M.);; 2Immunology Laboratory (LICE-OI), CHU Bellepierre, Reunion University Hospital, 97400 Saint-Denis, La Réunion, France; 3Rheumatology Clinical Department, CHU Bellepierre, Reunion University Hospital, 97400 Saint-Denis, La Réunion, France

**Keywords:** human synovial tissue-derived mesenchymal stem cells, immunoregulators, arthritis, arthritogenic alphaviruses

## Abstract

Infection by arthritogenic alphaviruses (aavs) can lead to reactive arthritis, which is characterized by inflammation and persistence of the virus; however, its mechanisms remain ill-characterized. Intriguingly, it has been shown that viral persistence still takes place in spite of robust innate and adaptive immune responses, characterized notably by the infiltration of macrophages (sources of TNF-alpha) as well as T/NK cells (sources of IFN-gamma) in the infected joint. Aavs are known to target mesenchymal stem cells (MSCs) in the synovium, and we herein tested the hypothesis that the infection of MSCs may promote the expression of immunoregulators to skew the anti-viral cellular immune responses. We compared the regulated expression via human synovial MSCs of pro-inflammatory mediators (e.g., IL-1β, IL6, CCL2, miR-221-3p) to that of immunoregulators (e.g., IDO, TSG6, GAS6, miR146a-5p). We used human synovial tissue-derived MSCs which were infected with O’Nyong-Nyong alphavirus (ONNV, class II aav) alone, or combined with recombinant human TNF-α or IFN-γ, to mimic the clinical settings. We confirmed via qPCR and immunofluorescence that ONNV infected human synovial tissue-derived MSCs. Interestingly, ONNV alone did not regulate the expression of pro-inflammatory mediators. In contrast, IDO, TSG6, and GAS6 mRNA expression were increased in response to ONNV infection alone, but particularly when combined with both recombinant cytokines. ONNV infection equally decreased miR-146a-5p and miR-221-3p in the untreated cells and abrogated the stimulatory activity of the recombinant TNF-α but not the IFN-gamma. Our study argues for a major immunoregulatory phenotype of MSCs infected with ONNV which may favor virus persistence in the inflamed joint.

## 1. Introduction

Alphaviruses are enveloped positive single-strand RNA viruses belonging to the *Togaviridae* family, transmitted by mosquitoes [1,2]. Infection by alphaviruses can lead to polyarthralgia and polyarthritis, and to chronic tissue inflammation of the joints. ‘Old World’ alphaviruses are generally associated with rheumatic diseases. The arthritogenic alphaviruses (aavs) are Chikungunya virus (CHIKV), O’Nyong-Nyong virus (ONNV), Ross River Virus (RRV), Barmah Forest virus (BFV), and Mayaro virus (MAYV) [3,4]. 

Acute infection by aav leads to a brief viremia (5 to 7 days long) [5], mainly under the control of antiviral type I interferon (IFN)/IFN-stimulated gene (ISG, e.g., ISG15) responses [6,7] and virus-neutralizing antibodies [8]. Patients with acute symptoms will first present with fever, polyarthralgia, polyarthritis, myalgia, and rash [9]. These symptoms may be due to the robust pro-inflammatory immune responses associated with widespread activation of monocyte/macrophages, dendritic cells, mesenchymal stem cells (MSC), endothelial cells, muscles cells, periosteum and keratinocytes [5]. Indeed, during the acute phase of infection by aav, pro-inflammatory cytokines such as IL-1 (interleukin 1), IL-6 (interleukin 6), TNF-α (Tumor necrosis- factor alpha), IFN-γ (Interferon gamma), and chemokines such as CCL2 (Chemokine Ligand 2), CXCL8 (Chemokine C-X-C motif ligand 8), and prostaglandins are abundantly expressed [10,11,12,13,14,15]. This production of pro-inflammatory cytokines and chemokines further favors the recruitment of monocytes/macrophage NK and T/B cells at the site of inflammation and results in synovial tissue inflammation and bone/cartilage destruction [1]. 

The persistent joint inflammation in patients with arthritis post-aav infection has been linked to the persistence of aav remnants (e.g., viral RNA) in the joint several months to years after the initial infection [5,16]. This viral persistence might be due to the capacity of aavs to repress the type I IFN antiviral response. Indeed, CHIKV has been shown to inhibit the expression of ISGs implicated in viral clearance by interfering with the STAT signaling pathway [17,18]. 

In the synovial joint, mesenchymal stem cells (MSCs) are the main target of aavs such as CHIKV [19,20]. Aavs can infect MSCs through the recently identified receptor matrix remodeling associated 8 (MXRA8) [21]. MSCs are clearly described as critical modulators of the immune system. They are positioned at the perivascular level (a portal of entry for infectious agents). MSCs express pattern recognition receptors (PRRs) such as Toll-like receptors (TLRs) to drive signals, leading to either a pro-inflammatory or anti-inflammatory phenotype of MSC depending on the microenvironment. In the presence of damage-associated signals (e.g., alarmin HMGB1) and/or pathogen-associated molecular patterns (PAMPs, e.g., viral ssRNA), quiescent MSCs can be activated. They adopt a pro-inflammatory phenotype and produce pro-inflammatory factors, e.g., IL-6 and CXCL-8. The pro-inflammatory phenotype of MSC is also characterized by low levels of indoleamine 2,3-dioxygenase (IDO). Conversely, MSCs can also be activated by TNF-α and IFN-γ to adopt an anti-inflammatory phenotype, leading this time to a reduction in inflammation through the robust expression of immunoregulators such as IDO [22], TNF-α-stimulated gene 6 (TSG6) [23], and canonical anti-inflammatory cytokines such as IL-10 and TGFβ [22].

IDO is a catabolic enzyme that degrades tryptophan along the kynurenine pathway. IDO can control local tissue inflammation, autoimmunity, and suppressed immunity in cancer and chronic infections [24]. Therefore, the immunoregulatory effect of IDO contributes to immune tolerance [25]. 

TSG6 also act as an anti-inflammatory mediator. TSG6 suppresses inflammation by inhibiting neutrophil migration and the production of pro-inflammatory cytokines via macrophages in different models of inflammation such as arthritis [26,27]. Indeed, TSG6 binds to chemokines such as CXCL8 though their glycosaminoglycan-binding sites, leading to the inhibition of neutrophil recruitment [27]. TSG6 also binds to other chemokines, such as CCL2 and CCL5, involved in pro-inflammatory activities. TSG6 can also act as an anti-inflammatory mediator through a complex with hyaluronan binding to CD44 macrophages. This binding leads to a decrease in TLR/NFκB signaling [28].

Two other immunoregulators have been described to be involved in inflammation: growth arrest specific 6 (GAS6) and protein S (PROS1). GAS6 and PROS1 are ligands of TAM (Tyro3, Axl and Mer) receptors known to inhibit the anti-inflammatory response [29,30,31]. Furthermore, injection of GAS6 and PROS1 in collagen-induced arthritis mice decreased the production of cytokines such as IL-12, IL-23, and IFN-gamma (IFN-γ). The injection of PROS1 also ameliorates arthritis [32].

In arthritis (e.g., RA), the inflammatory phenotype of synovial tissue-derived MSCs is known to be controlled with epigenetic modifications and involve different immunoregulatory micro-RNAs (mi-RNAs) [33]. miR-146a-5p is of particular interest and participates in the regulation of the lymphocyte T reg phenotype. Moreover, it has been shown that the upregulation of miR-146a in synovial tissue-derived MSCs decreases the production of IL-1β and IL-6. Interestingly, miR-146a expression was upregulated by pro-inflammatory mediators such as TNF-α, IL-1β, and IFN-γ [34]. In contrast, miR-221 has been described to stimulate inflammation. Indeed, the inhibition of miR-221 in human synovial tissue-derived MSCs stimulated with LPS decreased TNF-α, IL-1β, and IL-6 expression [35]. Moreover, miR-221-3p favored inflammation by downregulating the expression of anti-inflammatory IL-10 via M2 macrophages. miR-221-3p also favored M2 macrophages to produced pro-inflammatory cytokines [36].

The aim of this study was to determine if aavs (e.g., ONNV, a class II aav) alone or in the presence of a pro-inflammatory environment can modulate the expression of immunoregulators in human synovial tissue-derived MSCs. We demonstrated that ONNV, particularly when combined with TNF-α or IFN-γ, increased the expression of TSG6 and IDO. These results shed new light on the capacity of aavs such as ONNV to skew the anti-viral innate cellular immune response and to possibly grant viral persistence in synovial joints. 

## 2. Results

### 2.1. ONNV Can Infect Human Synovial Tissue-Derived MSCs

First of all, in order to ascertain whether ONNV can readily infect commercially available human synovial tissue-derived MSCs, cells were infected with ONNV (at a low MOI of 1) over 6 h or 24 h; then, qPCR and immunofluorescence were performed. The relative RNA expression of E1, E2, and capsid were higher at 6 h and 24 h compared to mock-infected control cells (Figure 1A). At 6 h, the fold changes for ONNV E1, E2, and capsid were 46,460-fold, 3212-fold and 112-fold, respectively. At 24 h, the fold changes were 15,161,461-fold, 2,183,901-fold, and 113,356-fold, respectively. The expression of NSP2 (non-structural protein 2) RNA was also increased in cells infected with ONNV at 6 h and 24 h (Appendix A). Moreover, the infection of human synovial tissue-derived MSCs with ONNV was confirmed via immunofluorescence using a specific rabbit anti-capsid antibody. Figure 1B shows that the majority of MSCs were infected at 6 h and more efficiently at 24 h.

### 2.2. ONNV (MOI of 1) Alone or in Combination with Pro-Inflammatory Mediators Was Not Cytotoxic to Human Synovial MSCs and Did Not Affect Cell Proliferation

The activity of all our treatments was evaluated on cell viability via a non-radioactive LDH release assay. As shown in Figure 2A, no treatments had a significant effect on LDH release and, hence, all were non-cytotoxic. A basal LDH release was detected in the control cells (untreated cells). Moreover, no treatments affected the mitochondrial activity of the human synovial tissue-derived MSCs (Figure 2B). Figure 2C shows that there was no significant difference in positive staining for the cell proliferation protein marker Ki67 between the different groups.

### 2.3. ONNV (MOI of 1) Alone or in Combination with Pro-Inflammatory Cytokines Promoted the Expression of the Antiviral ISG15 mRNA

Next, we analyzed the IFN type I-dependent ISG response after the infection of human synovial MSCs with a low MOI of ONNV, alone or in combination with TNF-alpha or IFN-γ. We tested this using qPCR *ISG15* (interferon stimulated gene 15) mRNA expression. The ONNV alone increased *ISG15* mRNA expression 24 h post-infection (15-fold). Moreover, in human synovial tissue-derived MSCs treated with TNF-α or IFN-γ and infected with ONNV, *ISG15* expression was further increased (at both time points) when compared to TNF-α alone or IFN-γ alone (Figure 3). 

### 2.4. ONNV (MOI of 1) Alone Failed to Upregulate the Expression of Pro-Inflammatory Factors (IL1β, IL6, CCL2)

To determine the effect of the regulatory activity of ONNV infection on the expression of pro-inflammatory mediators, we assessed, via qPCR, the levels of *IL-1β*, *IL6* and *CCL2* mRNAs in human synovial tissue-derived MSCs (Figure 4). The cells were either only infected with ONNV or both stimulated with TNF-alpha/IFN-gamma and infected. ONNV alone had no effect on *IL-1β* (Figure 4A), *IL6* (Figure 4B), and *CCL2* (Figure 4C) mRNA expression at 6 h and 24 h, when compared to mock-infected control cells. However, when human synovial tissue-derived MSCs were treated with TNF-α and infected with ONNV, the mRNA expression of *IL-1β* (up to 2-fold at 6 h and 24 h), *IL6* (up to 9-fold at 24 h), and *CCL2* (up to 2-fold at 6 h and up to 9-fold at 24 h) were increased when compared to TNF treatment alone. The same results were observed in human synovial tissue-derived MSCs treated with IFN-γ.

### 2.5. ONNV Alone and/or in the Presence of Pro-Inflammatory Mediators Can Upregulate the Expression of Immunoregulatory Factors IDO and TSG6 by Human Synovial MSCs 

qPCR was performed to determine the effects of ONNV alone or in the presence of IFN-γ/TNF-α on *IDO* and *TSG6* expression. IFN-γ is a major regulator of *IDO* expression [37]. In our experimental conditions, we confirmed that IFN-γ treatment increased *IDO* mRNA relative expression at 6 h (up to 3271-fold) and 24 h (up to 4950-fold), compared to the control (untreated cells). Moreover, ONNV increased *IDO* mRNA expression at 6 h (up to 6-fold) but had no effect on *IDO* expression at 24 h when compared to the control cells. Interestingly, when MSCs were both stimulated with IFN-γ and infected with ONNV, the mRNA relative expression of *IDO* was upregulated when compared to MSCs stimulated with IFN-γ alone at 6 h and 24 h (up to 1.14-fold and 1.18-fold, respectively) (Figure 5A). In order to determine the effect of ONNV on *TSG6* expression, synovial tissue-derived MSCs were stimulated with TNF-α, or infected with ONNV, or both stimulated and infected. As expected, *TSG6* mRNA expression was upregulated in the MSCs stimulated with TNF-α at 6 h and 24 h (up to 42-fold and 6-fold, respectively) compared to the control cells. Furthermore, ONNV only increased *TSG6* expression at 24 h, with a fold increase of four compared to the control. Interestingly, the ONNV in combination with TNF-α increased *TSG6* mRNA expression at 24 h, compared to the MSCs stimulated with TNF-α alone, up to 2.65-fold (Figure 5B).

### 2.6. ONNV Infection Increases GAS6 Expression but Has No Effect on PROS1 Expression in Human Synovial Tissue-Derived MSCs

We also determined via qPCR the effect of ONNV on two other immunoregulators. As shown in Figure 6, ONNV increased *GAS6* expression at 6 h (up to 2-fold) but had no significant effect at 24 h. *GAS6* expression also increased (up to 2-fold) in the cells stimulated with TNF-α and infected with ONNV as compared to the control (unstimulated cells). ONNV had no effect on *PROS1* mRNA expression at 6 h and 24 h. *PROS1* expression decreased in TNF-α stimulated cells as compared to the unstimulated cells. It should be noted that *PROS1* relative expression was very weak in the synovial tissue-derived MSCs.

### 2.7. There Was No Correlation between Pro- and Anti-Inflammatory Responses

We analyzed the association between pro- and anti-inflammatory cytokines. Our results showed that there was no significant correlation between the pro-inflammatory gene *CCL2* and anti-inflammatory genes *IDO* or *TSG6* (Figure 7).

### 2.8. ONNV Infection Downregulated the Expression of Immunoregulatory miR-146a-5p in Human Synovial MSCs Treated with TNF-α

Next, we studied the regulation of two mi-RNAs in our experimental model (Figure 8). Infection with ONNV decreased miR-146a-5p (0.5-fold) and miR-221-3p (0.5-fold) expression at 6 h but had no effect at 24 h when compared to the control cells. The level of miR-146a-5p increased in response to either TNF-α (up to 4-fold at 6 h, up to 3-fold at 24 h) or IFN-γ at 6 h (up to 5-fold). TNF-α also increased miR-221-3p expression (2-fold) at 6 h. Interestingly, when the human synovial tissue-derived MSCs were treated with TNF-α and infected with ONNV, miR-146a-5p expression decreased (0.4-fold) compared to the TNF-α treated cells at 6 h. The same results were observed for miR-221-3p at 6 h (0.6-fold). 

## 3. Discussion

Herein, we studied the regulated expression of canonical pro- and anti-inflammatory factors expressed in the joint by MSCs in response to aav infection. While ONNV (a class II aav) was found to promote a pro-inflammatory state of activation in MSCs in vitro, we also described a unique propensity of the infection to drive an anti-inflammatory MSC phenotype which may contribute to virus persistence by skewing the immune system. 

We demonstrated that all of our treatments were not cytotoxic and did not affect mitochondrial activity and cell proliferation. We confirmed via qPCR and immunofluorescence that the human synovial tissue-derived MSCs were infected with ONNV at MOI 1. The cells were readily infected at 6 h, but the infection was more efficient at 24 h. We hypothesized that the persistence of aavs in the joint might favor inflammation through the recruitment of immune cells and contribute to hyperplastic synovial pannus formation, cartilage degradation, and eventually bone erosion, as observed in rheumatoid arthritis (RA) [20]. Therefore, we studied the regulation of pro-inflammatory mediators in untreated human synovial tissue-derived MSCs or those treated with two important canonical cytokines, TNF-α or IFN-γ, alone or with ONNV.

First, infection with ONNV alone had no effect on pro-inflammatory mediators, including *IL-1β*, *IL6*, and *CCL2*. However, when ONNV was combined with TNF-α (known to be produced by synovial macrophages), the mRNA expressions of *IL-1β*, *IL-6*, and *CCL2* were particularly upregulated when compared to TNF treatment alone. These results suggest that the presence of pro-inflammatory mediators in synovial joints might favor the ability of ONNV to induce inflammation through the mobilization of latently infected human synovial-derived tissue MSCs to produce further key cytokines and chemokines. ISG15 is a potent antiviral factor reported to participate in viral clearance. We showed that ONNV, alone or combined with TNF-α, increased *ISG15* expression and therefore possibly limits viral spreading in other tissues of the body.

Second, MSCs of the joint also have important anti-inflammatory and immunoregulatory activities, particularly in response to IFN-γ (produced by T and NK cells). Next, we studied the expression and regulation of IDO in our in vitro model of MSCs. We confirmed that in the presence of pro-inflammatory settings, mimicked either by chemical poly I:C (a viral RNA analog) or recombinant IL-1β, human synovial tissue-derived MSCs expressed high levels of *IDO* mRNA, as described in the literature (Appendix A) [38,39,40,41,42]. We also showed that IFN-γ markedly increased *IDO* expression, as described in the literature [40,41,42]. Interestingly, we found that ONNV increased *IDO* expression at 6 h but not at 24 h, suggesting negative feedback of IDO. Moreover, we demonstrated for the first time that in association with IFN-γ, ONNV increased further *IDO* mRNA expression compared to human synovial-derived tissue MSCs stimulated with IFN-γ alone. This upregulation of IDO might permit virus tolerance through the polarization of monocytes toward M2 macrophages and the suppression of T cell responses [38,43,44]. Our observations agree with other studies, indicating that several viruses involved in arthritic diseases can escape immune control through IDO signaling pathways. Indeed, it was found that in several viral infections, such as human immunodeficiency virus, hepatitis B virus, hepatitis C virus, herpes, and cytomegalovirus, IDO expression was upregulated in antigen-presenting cells, leading to tolerance of the disease [45]. Moreover, tryptophan depletion detected by the GCN2 (general control nonderepressible 2) kinase in T lymphocytes, can lead to the initiation of a stress response program including cell cycle arrest and differentiation. Therefore, the suppression of mTOR signaling pathways and promotion of GCN2 kinase in T lymphocytes can lead to cell arrest and the inhibition of T cell proliferation. We have found that T cells are still infiltrating the synovial tissue of patients with arthritis months after CHIKV infection [46]. It is therefore important to stain tissue sections for IDO and T cell markers (i.e., transcription factors to identify Th1/Th17 versus Th2/Tregs), specifically in areas of the synovial tissue where the virus is present.

Third, we have studied the regulation of *TSG6* expression by pro-inflammatory mediators and ONNV. First, as described in the literature, we confirmed that *TSG6* was upregulated in response to a pro-inflammatory environment mimicked by poly I:C, IL-1β, and IFN-γ (Appendix A) [28,38,39,47,48]. In addition, as expected, we found that *TSG6* mRNA expression was upregulated in MSCs at 6 h and 24 h post-stimulation with TNF-α [28,47,48]. TSG6 has been detected in the synovial fluid of patients with arthritis [48]. It has been demonstrated that TSG6 is locally produced in the synovium and cartilage of arthritis patients [49]. Interestingly, herein, we demonstrated that ONNV alone increased *TSG6* mRNA expression (at 24 h). Moreover, ONNV potentialized the effect of TNF-α on *TSG6* expression. Therefore, as for IDO, we propose that aavs such as ONNV might persist in human synovial tissue-derived MSCs through the expression of TSG6, known to be a potent anti-inflammatory factor. TSG6 can be defined as a negative feedback regulator of inflammation. Moreover, TSG6 can bind to hyaluronic acid (HA), a ligand of CD44. The activation of CD44 by HA contributes to leukocyte migration during inflammation. While TSG6 binds to HA, TSG6 can block the HA–CD44 interaction [23]. In collagen-induced arthritis in mice, treatment with recombinant TSG6 ameliorated arthritis through decreasing pannus formation and cartilage erosion. In transgenic mice expressing TSG6, arthritis induced by type II collagen was alleviated [50]. In proteoglycan-induced arthritis, intra-articular injections of recombinant TSG6 prevented cartilage degradation [51]. TSG6 can also modulate the expression of GAS6, a ligand of TAM receptors and another immunoregulator involved in inflammation. GAS6 and PROS1 decrease cytokine production in synovium [32]. To our knowledge, the mechanics of the expression of *GAS6* from synovial MSCs was unknown, hence we next tested whether GAS6/PROS1 were expressed and regulated in human synovial tissue-derived MSCs. 

*GAS6* mRNA was decreased in the MSCs stimulated with recombinant IL-1β, TNF-α, TGFβ1 (produced by MSC), and PDGBB (produced by endothelial cells) at 6 h, while poly I:C, known to mimic viral infections, increased *GAS6* expression. *PROS1* mRNA was weakly expressed in unstimulated (control) and stimulated cells. We found that poly I:C, TNF-α, and PDGFBB decreased, while IFN-γ and TGFβ1 increased, *PROS1* mRNA expression (Appendix A). We found that ONNV alone increased *GAS6* expression (at 24 h); however, it had no effect on *PROS1* expression. 

Moreover, we showed that there was no correlation between the pro- and anti-inflammatory responses for each condition, suggesting that the anti-inflammatory response did not affect the pro-inflammatory response. 

Next, we studied the regulation of two mi-RNAs involved in joint inflammation: miR-146a-5p and miR-221-3p. miR-146a-5p has a protective role in arthritis. The upregulation of miR-146a-5p reduced the production of pro-inflammatory mediators (IL-1β, IL-6, and CXCL-8) by synovial tissue-derived MSCs [52]. On the contrary, miR-221 promotes inflammation, since its inhibition in LPS-stimulated synovial tissue-derived MSCs can result in the inhibition of TNF-α, IL-6, and IL-1β [35]. Unexpectedly, we showed that ONNV infection decreased miR-146a-5p and miR-221-3p expression equally. This was surprising to us, given that CHIKV was shown to increase miR-146a expression in synovial MSCs and that the upregulation of miR-146a favored virus replication [53]. We will need to explore further the regulation of miR-146a-5p expression by MSCs. We confirmed that the expression of this major immunoregulatory mi-RNA was increased in human synovial tissue-derived MSCs stimulated with TNF-α (but not IFN-γ), as described in the literature [34,54]. Interestingly, we showed for the first time that ONNV infection decreased miR-146a-5p expression in cells stimulated by TNF-α. miR-221-3p was more abundantly expressed (relative expression) by the unstimulated human synovial tissue-derived MSCs, and its expression was also similarly regulated by TNF alone or TNF with ONNV. It has already been established that mRNA expression was less elevated at 24 h than at 6 h. However, herein, the mRNA expression of miR-146a-5p was equivalent at 6 h and 24 h. We therefore suggest that the miR-146a-5p was partially subject to self-regulation.

The strength of our study is that we have equally considered how an alphavirus such as ONNV, as a model of arthritis post-aav infection, may on one hand mobilize a full immune response (innate and adaptive, producing several cytokines) and yet favor viral persistence. Importantly, we found that the activities of critical immune factors such as TNF-α and IFN-γ (produced by newly recruited monocytes/macrophages/T lymphocytes) equally drive the expression of pro-inflammatory, as well as anti-inflammatory, activities in the infected joint. 

One of the limitations of our study is that we have only studied the immunoregulatory activities at transcriptional levels. Further studies are required in order to confirm the effects of ONNV on IDO, TSG6, and GAS6 at the protein levels in vitro and in the aforementioned tissue biopsies. Antibodies will need to be selected and tested on paraffin wax sections in human and animal models. From a clinical standpoint, immunoregulators such as IDO, TSG6, and GAS6 might represent therapeutic targets for arthritis post-aav infection. Moreover, it is important to note that the MSCs did not only secrete IDO, TSG6, and GAS6; they also secreted other various factors and growth factors involved in the regulation of inflammation. Further studies are necessary to understand the immunoregulatory role of MSCs.

## 4. Materials and Methods

### 4.1. Cell Culture Reagents and Viruses

Human synovial tissue-derived MSCs were purchased from ScienCell research laboratories (ScienCell, 4700, Clinisciences, Carlsbad, CA, USA). Cells were cultured with Modified Eagle Medium (MEM, PAN Biotech, P0408500, Aidenbach, Germany) and supplemented with 10% of decomplemented fetal bovine serum (FBS, PAN Biotech, 3302 P290907, Aidenbach, Germany), 1 mM of sodium pyruvate (PAN Biotech, P0443100, Germany), 2 mM of L-glutamine (Biochrom AG, K0282, Berlin, Germany), 0.1 mg/mL of penicillin-streptomycin (PAN Biotech, P0607100), and 0.5 µg/mL of amphotericin B (PAN Biotech, P0601001). Cells were maintained at 37 °C with 5% of CO_2_. Cells were used up to passage 9. ONNV was obtained from the alphavirus national reference center of Marseille (CNR, Marseille, France). 

### 4.2. Cell Treatments

Human synovial tissue-derived MSCs were placed in 6-well plates or in coverslips and were maintained at 37 °C in a humid atmosphere with 5% of CO_2_. The medium was replaced twice a week. MSCs were stimulated with TNF-α or IFN-γ at 20 ng/mL for two hours. Then, ONNV at MOI 1 (multiplicity of infection 1) was added in the media containing cytokines over 6 h or 24 h at 37 °C, in a humid atmosphere with 5% of CO_2_. Synovial tissue-derived MSCs were also infected with ONNV alone over 6 h or 24 h. For cell infection with ONNV at MOI 1, the cells were counted and one virus was added per one cell using virus stock at 10^7^ virus/mL.

### 4.3. Cytotoxicity Assay

Cytotoxicity assay was performed via quantitative release of lactate dehydrogenase (LDH) from damaged cells (ref. G1781, CytoTox96 Non-Radioactive Cytotoxicity Assay, Promega, Madison, WI, USA). Human synovial tissue-derived MSCs were grown in a 96-well plate and stimulated with TNF-α or IFN-γ for two hours at 20 ng/mL, and then infected with ONNV at MOI 1 over 24 h. Cells were also infected with ONNV alone over 24 h. Measurement of LDH release was performed following the manufacturer’s instructions. The released LDH in the supernatant after treatment was compared to the maximum LDH release (intracellular LDH induced via the addition of Triton 1%). 

### 4.4. Mitochondrial Metabolic Activity Measurements

Human synovial tissue-derived MSCs were grown in a 96-well plate and stimulated with TNF-α or IFN-γ for two hours at 20 ng/mL and then infected with ONNV at MOI 1 over 24 h. Cells were also infected with ONNV alone over 24 h. A total of 4 h before the end of the experiment, 20 µL of sterile-filtered MTT (3-(4-5-dimethylthiazol-2-yl)-2,5-diphenyltetrazolium bromide) solution (5 mg/mL) (Sigma-Aldrich, Darmstadt, Germany) was added to each well and the plate was incubated at 37 °C. Then, the plate was centrifugated to remove the unreacted dye. Formazan crystals were dissolved by adding 200 µL of dimethyl sulfoxide. Absorbance was measured at 560 nm (Biotek Cytation 5 imaging reader, Winooski, VT, USA). 

### 4.5. RNA Extraction and qPCR

Total RNA from human synovial tissue-derived MSCs stimulated with TNF-α or IFN-γ (20 ng/mL) and infected with ONNV (MOI 1), or only infected with ONNV (MOI 1), was extracted directly from the harvested cell culture in a 6-well plate using a Quick-RNA^TM^ viral kit (Zymo, Ozyme Irvine, CA, USA, Cat no. R1035). qPCR experiments were performed using the SensiFast Probe No-ROX One-Step Kit (Meridian Bioscience, Bioline, London, UK, Cat no. BIO-76005), to which Sybr green was added beforehand. qPCR tests were executed in a Quant studio 5 PCR thermocycler (Thermo Fisher Scientific, Waltham, MA, USA) according to the following steps: reverse transcription at 42 °C for 5 min and 40 cycles, including a denaturation step at 95 °C for 5 s, annealing step at 58 °C for 15 s, and extension step at 72 °C for 15 s. Fluorescence data were obtained at 520 nm during the extension step. GAPDH was used as a housekeeping gene. Samples were analyzed in three independent experiments. The sequences of the primers used are available in Table 1. 

### 4.6. mi-RNA Extraction and RT-PCR

Human synovial tissue-derived MSCs were treated with either recombinant TNF-α, IFN-γ (both 20 ng/mL), or ONNV (MOI 1), alone or when combined (cytokine with ONNV). mi-RNAs were extracted directly from the harvested cell culture in a 6-well plate using miRNeasy Serum/Plasma Advanced Kit (Qiagen, Hilden, Germany, ref. 217204). Before mi-RNA extraction, 250 µL of NaCl (0.09%) was added to each well, collected, and kept at −80 °C until use. Reverse transcription (miScript II RT, ref. 218161) was performed in a final volume of 20 µL (5 µL of RNA and 15 µL of enzyme mix). cDNA was collected and conserved at −20 °C until use. qPCR (miScript Sybr green PCR, Qiagen, ref. 218075) was performed in a final volume of 5 µL (1 µL of extracted cDNA per reaction, 3 µL of enzyme mix, and 1 µL of primer mix, with a final primer concentration of 1.25 µM). qPCR testing was executed in a Quant studio 5 PCR thermocycler (Thermo Fisher Scientific). Relative gene expression was established using Ce-39 (Qiagen, ref. MS00019789). The primers used for mi-RNA analyses are listed in Table 2.

### 4.7. Immunofluorescence

Human synovial tissue-derived MSCs were grown on sterile glass coverslips in a 24-well plate and then infected with ONNV at MOI 1 for 6 h, or stimulated with TNF-α or IFN-γ (20 ng/mL) and infected with ONNV (MOI 1) for 6 h or 24 h, at 37 °C in a humid atmosphere with 5% of CO_2_. Coverslips were washed three times in PBS and then fixed in cold 99% ethanol at room temperature for ten minutes. Cells were incubated with the primary antibody against alphavirus, which was either capsid or rabbit anti-Ki67, overnight at 4 °C. Coverslips were washed three times with PBS/BSA 1% and two times with distilled water. Then, they were incubated for two hours with the secondary antibody Alexa Fluor^®^ 594-conjugated donkey anti-rabbit (Invitrogen, Thermo Fisher Scientific, Waltham, MA, USA). The nuclei were revealed via staining with nuclear fluorochrome 4′,6-diamidino-2-phenylindole (DAPI). After washing with PBS/BSA 1% and distilled water, the coverslips were mounted with vectashield (Vector Labs, Newark, CA, USA). Staining was visualized under a Nikon Eclipse E2000-U microscope. Images were captured and processed using a Hamamatsu ORCA-ER camera and the NIS-Element BR 5.20.00 imaging software (Nikon, Tokyo, Japan). Magnification was ×600. The Ki67 mean fluorescence intensity was determined using ImageJ software v1.54d.

### 4.8. Statistical Analysis

All assays were performed over three independent experiments. Data were expressed as mean ± standard error of the mean (SEM). Statistical analysis was carried out using Graph Prism 8 software. Significant differences between the means were determined via analysis of variance (two-way ANOVA) procedures, followed by a multiple comparison test (Bonferroni test). A Spearman test was also used to determine the correlation between the pro- and anti-inflammatory responses.

## 5. Conclusions

We observed that human synovial tissue-derived MSCs expressed immunoregulators including IDO and TSG6 in response to stimulation with TNF-α and IFN-γ, used alone or in combination with ONNV to mimic clinical settings. We used a class II aav to readily test for infection and to show that infected human synovial tissue-derived MSCs, through the upregulated expression of IDO, TSG6, and GAS6, may attenuate inflammation and favor immune tolerance (Figure 9).

## Figures and Tables

**Figure 1 ijms-24-15932-f001:**
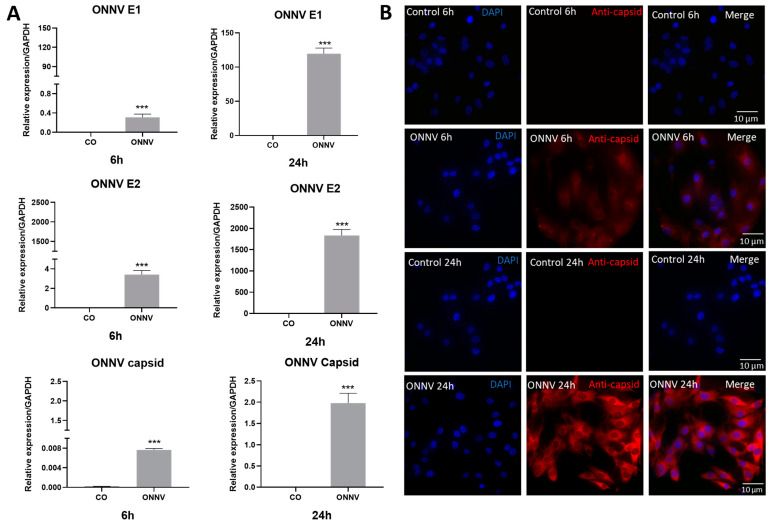
Synovial tissue-derived MSCs were readily infected with ONNV. Human synovial tissue-derived MSCs were infected with ONNV at MOI 1 over 6 h or 24 h. (**A**) Total RNA was collected and the relative viral RNA expressions of ONNV E1, ONNV E2, and ONNV capsid were determined by qPCR (Sybr green technique). The expression was calculated by Delta Ct relative to the housekeeping gene (GAPDH). Reported values are means ± SEM of three independent experiments, and *p*-values were calculated using an unpaired *t* test: *** *p* < 0.001 as compared to control (uninfected cells). (**B**) Human synovial MSCs were probed with primary antibody rabbit anti-ONNV capsid (Dr A Merits, Estonia) and nuclei were stained with DAPI (blue). Next, an Alexa Fluor 594-conjugated donkey anti-rabbit (red) antibody was used (×60 oil objective).

**Figure 2 ijms-24-15932-f002:**
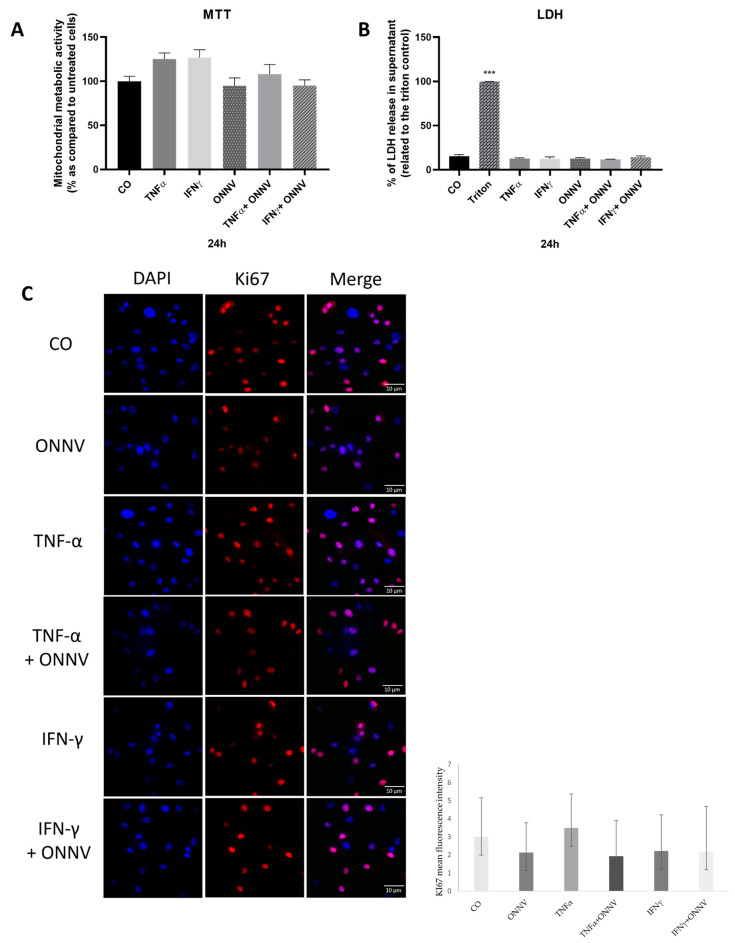
No treatments had a significant effect on LDH release, mitochondrial activity, and cell proliferation. Human synovial tissue-derived MSCs were stimulated with TNF-α and IFN-γ at 20 ng/mL, or infected with ONNV at MOI 1 alone or in the presence of TNF-α or IFN-γ. (**A**) The amount of released LDH in the culture medium was determined using CytoTox 96^®^ non-radioactive cytotoxicity assay and expressed relative to the maximum release (Triton 1%). (**B**) Mitochondrial metabolic activity was determined via MTT assay. (**C**) Cells were probed with primary antibody rabbit anti-Ki67 and nuclei were stained with DAPI (blue). Next, an Alexa Fluor 594-conjugated donkey anti-rabbit (red) antibody was used (×60 oil objective). Reported values are means ± SEM of three independent experiments, and *p*-value was calculated using an ANOVA test with Bonferroni multiple comparison: ***: *p* < 0.001 as compared to control (untreated cells).

**Figure 3 ijms-24-15932-f003:**
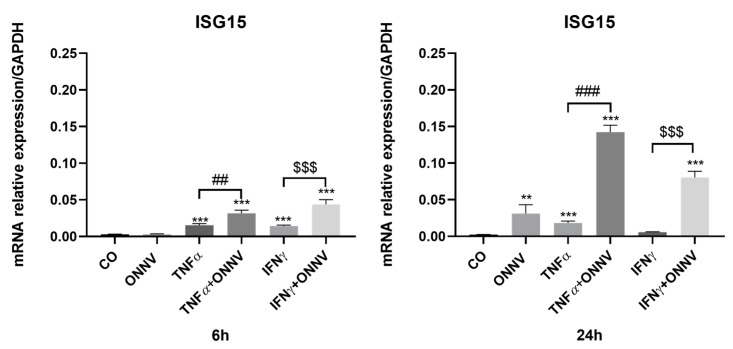
ONNV increased *ISG15* mRNA expression in untreated cells or cells treated with TNF-α or IFN-γ. Human synovial tissue-derived MSCs were stimulated with TNF-α or IFN-γ at 20 ng/mL and/or infected with ONNV at MOI 1 over 6 h or 24 h. qPCR was performed to determine the expression of *ISG15*. Reported values are means ± SEM of three independent experiments, and *p*-value was calculated using an ANOVA test with Bonferroni multiple comparison: **: *p* < 0.01, ***: *p* < 0.001 as compared to control; ##: *p* < 0.01, ###: *p* < 0.001 as compared to TNF-α alone; $$$: *p* < 0.001 as compared to IFN-γ alone.

**Figure 4 ijms-24-15932-f004:**
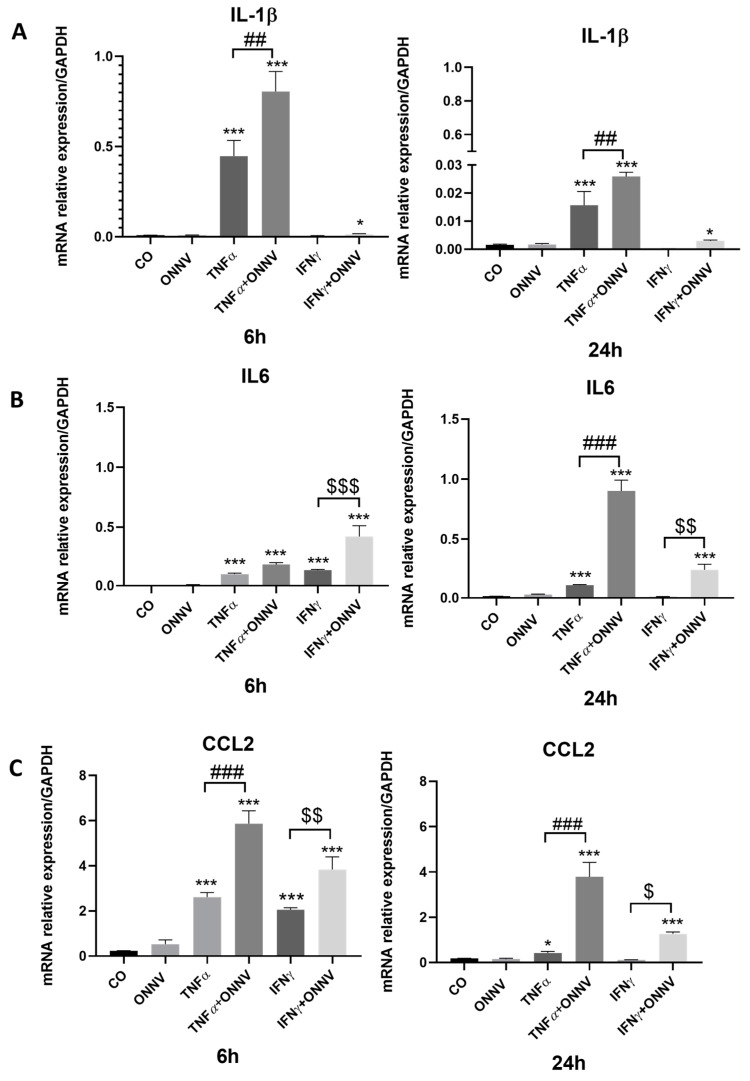
In the presence of TNF-α or IFN-γ, ONNV increased pro-inflammatory mediators. Human synovial tissue-derived MSCs were stimulated with TNF-α or IFN-γ at 20 ng/mL and/or infected with ONNV at MOI 1 over 6 h or 24 h. qPCR was performed to determine the expression of (**A**) *IL-1β*, (**B**) *IL6*, and (**C**) *CCL2* under the different conditions. Reported values are means ± SEM of three independent experiments, and *p*-value was calculated using an ANOVA test with Bonferroni multiple comparison: *: *p* < 0.05, ***: *p* < 0.001 as compared to control (untreated cells); ##: *p* < 0.01, ###: *p* < 0.001 as compared to TNF-α alone; $: *p* < 0.05, $$: *p* < 0.01; and $$$: *p* < 0.001 as compared to IFN-γ alone.

**Figure 5 ijms-24-15932-f005:**
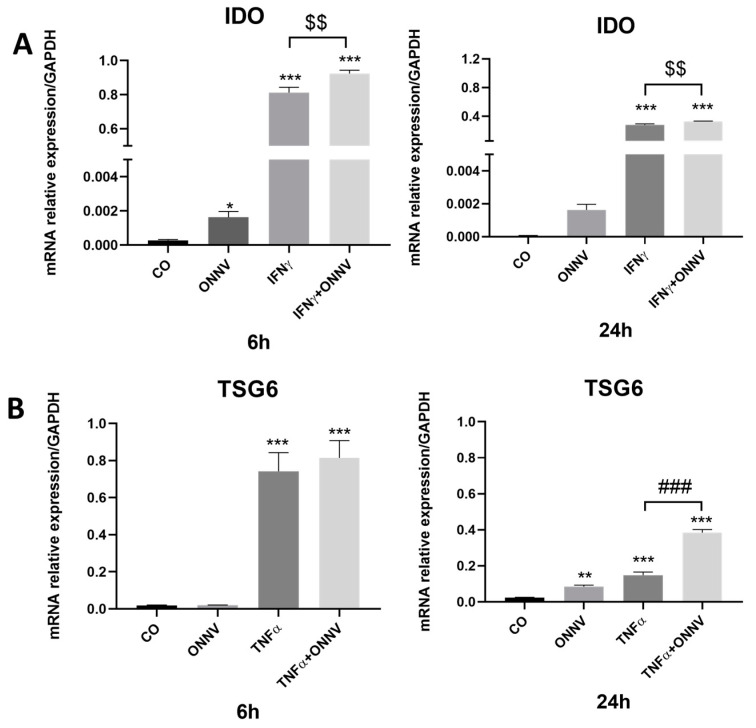
ONNV increased *IDO* and *TSG6* mRNAs expression in both untreated synovial tissue-derived MSCs and those treated with IFN-γ or TNF-α, respectively. (**A**) Human synovial tissue-derived MSCs were stimulated for 6 h or 24 h with IFN-γ at 20 ng/mL, or infected with ONNV at MOI 1, or infected with ONNV at MOI 1 in the presence of IFN-γ. (**B**) Cells were stimulated with TNF-α at 20 ng/mL, or infected with ONNV at MOI 1 alone or in the presence of TNF-α. qPCR was performed to determine the expression of *IDO* and *TSG6* under the different conditions. Reported values are means ± SEM of three independent experiments, and *p*-value was calculated using an ANOVA test with Bonferroni multiple comparison: *: *p* < 0.05, **: *p* < 0.01, ***: *p* < 0.001 as compared to control; ###: *p* < 0.001 as compared to TNF-α alone; $$: *p* < 0.01 as compared to IFN-γ alone.

**Figure 6 ijms-24-15932-f006:**
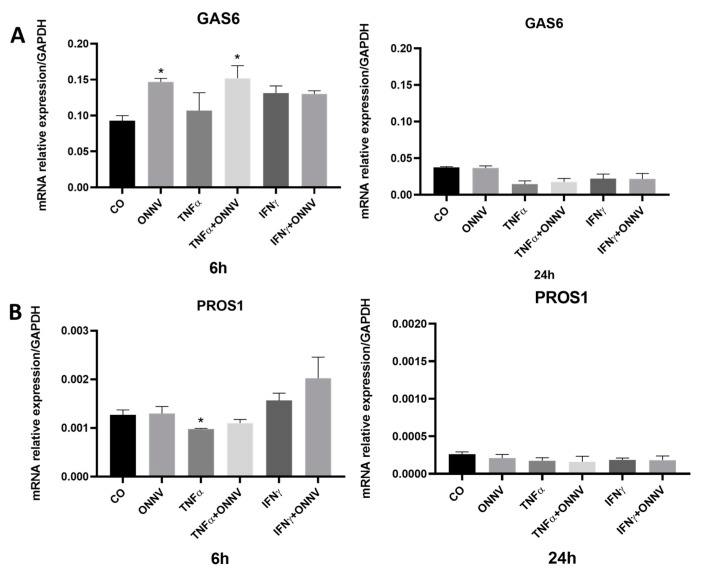
ONNV increased *GAS6* mRNA expression but had no effect on *PROS1* mRNA expression in synovial tissue-derived MSCs. Cells were infected with ONNV at MOI 1 over 6 h or 24 h. RNA was collected and the mRNA relative expression of *GAS6* (**A**) and *PROS1* (**B**) were determined by qPCR. Reported values are means ± SEM of three independent experiments, and *p*-values were calculated using an ANOVA test with Bonferroni multiple comparison: * *p* < 0.05 as compared to control (untreated cells).

**Figure 7 ijms-24-15932-f007:**
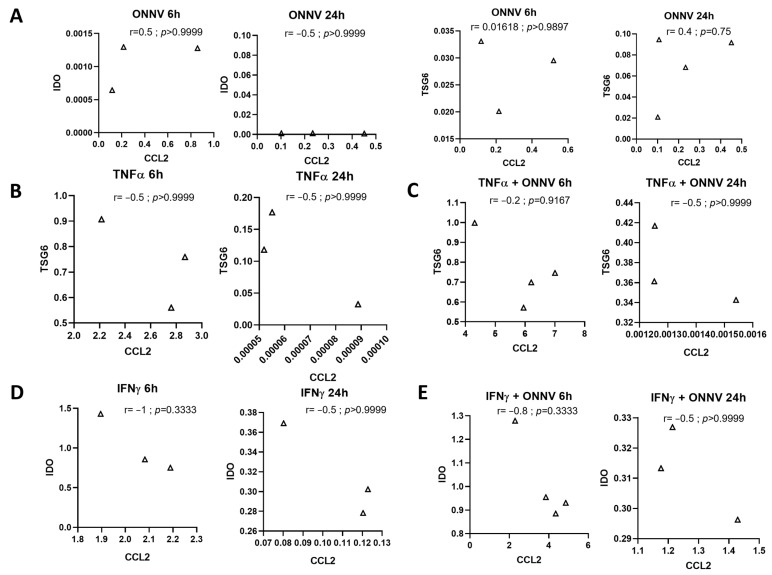
No correlation between pro- and anti-inflammatory responses in cells infected with ONNV alone (**A**), stimulated with TNF-α (**B**), stimulated with TNF-α and infected with ONNV (**C**), stimulated with IFN-γ alone (**D**), and stimulated with IFN-γ and infected with ONNV (**E**). *CCL2* has been used as pro-inflammatory gene, and *IDO* or *TSG6* have been used as anti-inflammatory genes. Statistical analysis was performed using a non-parametric Spearman test.

**Figure 8 ijms-24-15932-f008:**
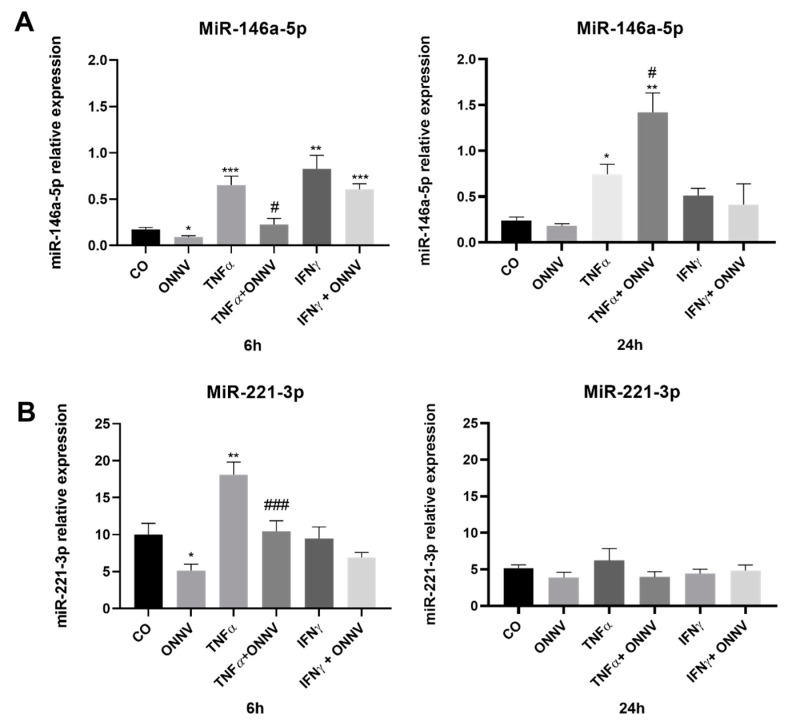
ONNV can downregulate the expression of miR-146a-5p (alone or in response to TNF). Synovial tissue-derived MSCs were stimulated with TNF-α or IFN-γ at 20 ng/mL, or infected with ONNV at MOI 1 alone or in the presence of TNF-α or IFN-γ. qPCR was performed to determine the expression of miR-146a-5p (**A**) and miR-221-3p (**B**) under the different conditions. Reported values are means ± SEM of three independent experiments, and *p*-value was calculated using an ANOVA test with Bonferroni multiple comparison: *: *p* < 0.05, **: *p* < 0.01, ***: *p* < 0.001 as compared to control; #: *p* < 0.05, ###: *p* < 0.001 as compared to TNF-α alone.

**Figure 9 ijms-24-15932-f009:**
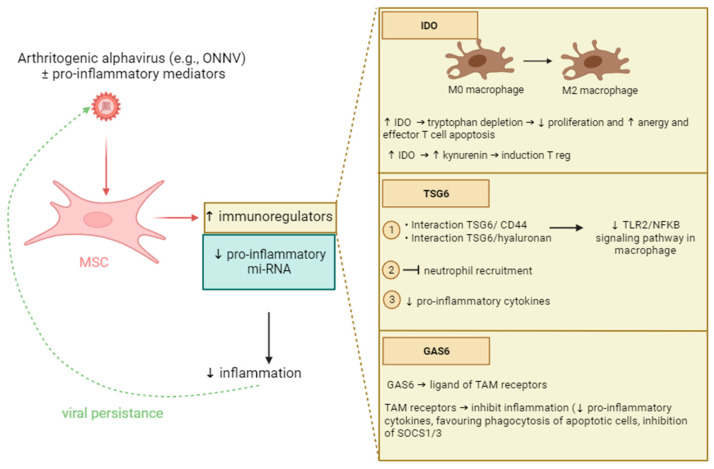
An alphavirus (e.g., Chikungunya class III or ONNV class II, which are easier to study) might persist in human synovial tissue-derived MSCs through its capacity to modulate the expression of several immunoregulators, particularly in the context of innate and adaptive immune cell responses (i.e., macrophages producing TNF-α and IFN-γ produced by recruited T cells). In our study, we found that pro-inflammatory mi-RNA (e.g., miR-221) expression by MSCs is also downregulated by ONNV.

**Table 1 ijms-24-15932-t001:** Sequence of primers used for qPCR analyses.

Target Gene	Forward Sequence (5′–3′)	Reverse Sequence (3′–5′)
GAPDH	TGCGTCGCCAGCCGAG	AGTTAAAAGCAGCCCTGGTGA
GAS6	ACGACCCCGAGACGGATTAT	CTTCCTATCGCAGGGGTTGG
IDO	AGCCCCTGACTTATGAGAACA	AGCTATTTCCAACAGCGCCT
MXRA8	TACTGTGGCCTGCACGAAC	CTCTCGGGGACGATGACATT
ONNV Capsid	TAGAACACGCCCGTCGTATG	GCAACGCCTTCAGAAACGC
ONNV E1	CACCGTCCCCGTACGTAAAA	GGCTCTGTAGGCTGATGCAA
ONNV E2	CCCCTGACTACACGCTGATG	CCTTCATTGGAGCCGTCACA
PROS1	GTCTCAGAGGCAAACTTTTGTT	AGAATTTGCACGACGCTTCC
TSG6	TGGCTTTGTGGGAAGATACYGT	TGGAAACCTCCAGCTGTCAC

**Table 2 ijms-24-15932-t002:** Sequences of mi-RNAs primers used.

Target mi-RNAs	Forward (5′–3′)
hsa-miR-146a-5p	GCAGTGAGAACTGAATTCCATG
hsa-miR-221-3p	GCAGAGCTACATTGTCTGCT

## Data Availability

On demand—EPI research unit.

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
