# Peer review of "Human Synovial Mesenchymal Stem Cells Expressed Immunoregulatory Factors IDO and TSG6 in a Context of Arthritis Mediated by Alphaviruses"

_ijms, 2023, doi:10.3390/ijms242115932_

Round 1
Reviewer 1 Report
Comments and Suggestions for Authors
Dear authors,
I'm afraid that the in vitro experiment did not mimic a chronic inflammatory process. To that, more than 24 hours would be necessary to evaluate a long term cell system alteration.
In addition, I could not find any citation on the literature for the miRs analyzed besides your review in 2021, which states that the samples were "human samples" and not specific MSCs. How do you explain that? Why do not analyze others miRs not connect with tumor development ?
Also, was the media containing TNF or IFN changed to a fresh one w/o cytokines before the virus addition? Was the virus added into a media with cytokines at any time point?
Why do not test MSCs function after the infection? Are they anti or proinflammatory after the infection?
I do not agree with the title as the cell culture was not performed to reproduce a chronic arthritis model.
Reviewer 2 Report
Comments and Suggestions for Authors
The study shows that in response to ONNV infection, the MSC produce pro-inflammatory but also regulatory response during chronic phase of inflammation. Resulting in incomplete clearance of the virus. However, the authors have used just 24 h culture to mimic chronic inflammatory milieu and show induction of immune regulatory response. Please provide the notion behind the planning of the experiment.
A correlation between increase in anti-inflammatory response with inflammatory response with time will give a better picture about virus infection mediated changes.
The data shows that both anti and pro-inflammatory response go down by 24 h except for expression of miR-146. Please discussion the context in the discussion.
Figure 2 please provide scale bar and quantify Ki67+ cells.
Line-252 TSG6?
Line-309 in TSG6
Line-318 de TSG6
Comments on the Quality of English LanguageMinor editing
Round 2
Reviewer 1 Report
Comments and Suggestions for Authors
Dear authors
I see and understand that you made all required modifications by both reviewers. No major comments from my side.